# Synthesis, Biological Evaluation, DNA Binding, and Molecular Docking of Hybrid 4,6-Dihydrazone Pyrimidine Derivatives as Antitumor Agents

**DOI:** 10.3390/molecules28010187

**Published:** 2022-12-26

**Authors:** Hairong Lan, Junying Song, Juan Yuan, Aiping Xing, Dai Zeng, Yating Hao, Zhenqiang Zhang, Shuying Feng

**Affiliations:** 1School of Pharmacy, Henan University of Chinese Medicine, Zhengzhou 450046, China; 2Academy of Chinese Medical Sciences, Henan University of Chinese Medicine, Zhengzhou 450046, China; 3Medical College, Henan University of Chinese Medicine, Zhengzhou 450046, China

**Keywords:** dihydrazone, pyrimidine, antitumor, apoptosis, DNA binding, molecular docking

## Abstract

In the present paper, on the basis of molecular hybridization, a series of 4,6-dihydrazone pyrimidine derivatives containing the pyridine moiety were synthesized, structurally characterized, and evaluated in vitro for their antitumor activity. According to the results, all the tested compounds demonstrated broad-spectrum antitumor activity against selected tumor cell lines (MCF-7, BGC-823, A549, and BEL-7402) and no obvious toxicity toward normal cells HL-7702. In particular, compounds **10a** and **10f** were found to be the most promising antitumor agents among the tested compounds against BGC-823 cells (IC_50_ = 9.00 μM and 7.89 μM) and BEL-7402 cells (IC_50_ = 6.70 μM and 7.66 μM), respectively. Compounds **10a** and **10f** exhibited higher potency against BGC-823 and BEL-7402 than the positive control 5-FU (IC_50_ = 15.18 μM and 15.81 μM). Further mechanism investigations demonstrated that compounds **10a** and **10f** could significantly increase the level of cellular ROS and induce early apoptosis of BGC-823 cells in a dose-dependent manner. Moreover, the DNA binding results from UV/Vis, CD spectroscopy, and molecular docking studies indicated that **10a** and **10f** bind with DNA via groove binding and partial intercalation. These results demonstrated that **10a** and **10f** may serve as novel lead compounds for the discovery of more dihydrazone pyrimidine derivatives with improved antitumor potency and selectivity.

## 1. Introduction

Tumor is a serious disease that endangers human life and health. At present, the treatment methods of malignant tumor mainly include surgery, radiotherapy, chemotherapy, and molecular targeted therapy, among which chemotherapy is one of the most effective methods. However, the current chemotherapeutic drugs generally have disadvantages, such as poor selectivity, strong drug resistance, and obvious side-effects; therefore, it remains a major challenge for researchers in developing novel antitumor agents with reduced toxicity and improved selectivity. Since the introduction of the first antimetabolic antitumor drug 5-fluorouracil (5-FU) [1] in 1957, a series of fluorouracil drugs have been developed successively, and the antitumor activity of pyrimidine derivatives has attracted more and more attention from researchers [2]. The pyrimidine moiety plays an important role in molecular design, synthesis, and bioactivity of new drugs because of its physiological activity [3,4,5,6]. According to literature reports, pyrimidine derivatives have been becoming important drug intermediates for the antitumor, bactericidal and antiviral activities [7,8,9,10]. 

Up to now, it has been reported that pyrimidine derivatives as potential antitumor agents mainly focused on pyrrolepyrimidine derivatives, thienopyrimidine derivatives, pyridopyrimidine derivatives, pyrazolopyrimidine derivatives, aminopyrimidine derivatives, etc. [11,12,13,14,15] (Figure 1). For example, pyrrolepyrimidine (**1**) was reported as an EGFR inhibitor to selectively inhibit the dynamic of HCC827 cells [11]. Thienopyrimidine (**2**) was reported as a dual epidermal growth factor receptor kinase and microtubule inhibitors (IC50 = 30 nM) [12]. Pyridopyrimidine (**3**) showed better antitumor activity against MCF-7 and MDA-MB-361 [13]. In addition, it was described that pyrazolopyrimidine (**4**) was a potent ATP-competitive inhibitor of the mammalian target of rapamycin (mTOR) [14]. Furthermore, 2,4-pyrimidinediamine (**5**) was reported as a potent dual inhibitor of ALK and HDAC [15], respectively. However, the antitumor activity of hydrazone pyrimidine derivatives has rarely been reported. To our knowledge, only a few monohydrazone derivatives, but no dihydrazone derivatives of pyrimidine have been studied so far [16,17] (Figure 1). Compound (**6**), a phenylhydrazone derivative of pyrimidine, can remarkably inhibit the proliferation of A549, H460, and HT-29 cells [16]. Compound (**7**), a dihydronaphthalen hydrazone derivative of pyrimidine, exhibited significantly potent anticancer activity against several tumor cell lines [17]. As a hydrogen bond donator and acceptor, hydrazone was prone to forming hydrogen bonds with a molecular target. On the other hand, the hydrazone functional group can improve the flexibility of a chemical structure to avoid steric hindrance, which might be beneficial to the biological activity and the solubility of compounds [18]. Furthermore, hydrazone can be as the linker to merge two or more functional groups together. Therefore, it is often used as an active building block in antiviral and antitumor small-molecule drugs [19,20,21,22,23]. The dihydrazone moiety can effectively increase the number of heterocycles in a molecule, which is expected to improve the antitumor activity of the molecule. The pyridine moiety has also been used as the core structure of many antitumor agents [24,25,26].

Herein, inspired by the versatility of the pyrimidine skeleton, hydrazone moiety, and pyridine functional group mentioned above, a series of 4,6-dihydrazone pyrimidine derivatives (**10a**–**10f**) (Figure 1) based on a merging approach were designed, synthesized, and evaluated for their antitumor activities. Among them, **10e** and its Hg(II) and Pb(II) complexes were previously reported in the literature [27]; however, the antitumor activity of compound **10e** has not been reported. Substituents at the R_1_ position were added to the terminal pyridine ring to investigate the electronic effect. In order to explore the effect of the molecular plane on antitumor activity, phenyl groups were also introduced at the R_2_ position of the pyrimidine skeleton for further modification. All target compounds were assayed for their antitumor activity in vitro against selected cell lines MCF-7, BGC-823, A549, BEL-7402, and HL-7702. On the basis of the antitumor results, promising compounds were selected for further mechanism studies. AO/EB double staining, flow cytometry, and ROS detection were carried out to confirm the apoptosis mechanism. The probable interaction mode of promising compounds with DNA was studied by UV/Vis, circular dichroism (CD) spectroscopy, and molecular docking.

## 2. Results and Discussion

### 2.1. Synthesis and General Characterization

The target compounds **10a**–**10f** were synthesized via a two-step reaction according to the synthetic routes depicted in Figure 1. First of all, intermediates **9a** and **9b** were synthesized by nucleophilic substitution reaction; secondly, **10a**–**10f** were synthesized by condensation reaction of the above intermediates and selected various pyridine aldehydes, respectively. Compounds **10a**–**10f** were all purified by column chromatography using the same eluent, with yields above 50%.

The chemical structures of **10a**–**10f** were fully identified by ^1^H NMR, ^13^C NMR, ESI-HRMS, IR, and UV/Vis spectra (Appendix A). Signals characteristic for the pyridine ring are multiple peaks at 8.61–7.16 ppm in the ^1^H NMR spectra. The presence of hydrazone group was confirmed by broad signal of NH protons and CH=N protons at 11.59–11.32 ppm and 8.27–8.15 ppm, respectively. For **10d**–**10f**, the presence of a benzene ring was supported by the chemical shift at 8.04–7.20 ppm in the ^1^H NMR spectra. Furthermore, for **10c** and **10f**, the presence of a methyl group was described by unimodal signal at 2.47 and 2.42 ppm in the ^1^H NMR spectra. For the ^13^C NMR analysis of **10a**–**10f**, the presence of hydrazone group was confirmed by broad signal of CH=N protons at 140.54–137.26 ppm. For **10c** and **10f**, the presence of a methyl group was supported by the chemical shift at 24.08 and 24.09 ppm, respectively. Compounds **10a**–**10f** are symmetric, and the results of ^1^H NMR and ^13^C NMR were consistent with their structures. In the ESI-HRMS analysis of **10a**–**10f**, the molecular ion peaks were observed as [M + H]^+^ for all the compounds.

The IR spectra of **10a**–**10f** showed similar absorption bands. The N–H stretching vibrations were located in the range of 3191–3299 cm^−1^ [28], and the C=N stretching vibrations were observed in the region of 1525–1598 cm^−1^. The C–C stretching vibrations occurred in the region of 1467–1386 cm^−1^. The absorption bands at 1450 cm^−1^ and 1454 cm^−1^ corresponded to the –CH_3_ stretching vibrations of **10c** and **10f**. In addition, for **10a**–**10f**, the observed absorption bands of 689–985 cm^−1^ were assigned to stretching vibration of benzene ring. Conventional UV/Vis characterization of **10a**–**10f** is shown in Appendix A. As we can see, different substituents had almost no effect on peak positions. The UV/vis spectra of **10a**–**10f** were similar, all displaying strong absorption peaks near 205 nm because of the existence of C=N functional groups and the big conjugate system in the structure. The absorption peak of C=N functional group is n→π*, which is assigned to the transition of lone pair electrons from unbonded heteroatoms to π* antibonding orbitals; the conjugate system is usually attributed to the π→π* transition overlap and benzene ring vibration. In summary, the absorption peaks near 205 nm of **10a**–**10f** were generated by n/π→π* transition.

Therefore, the successful synthesis and purity of **10a**–**10f** were fully characterized by ^1^H NMR, ^13^C NMR, ESI-HRMS, IR, and UV/Vis spectra.

### 2.2. Evaluation of Antitumor Activity In Vitro

#### 2.2.1. Analysis of Stability

The stability of a compound is crucial for the biological activity. The time-dependent UV/Vis spectra (Appendix A) of **10a**–**10f** in Tris-HCl buffer solution (pH 7.4) were recorded to evaluate the stability. As shown, the peak position and intensity of the strong absorption peaks near 205 nm in **10a**–**10f** did not change significantly within 48 h, which indicated that **10a**–**10f** were stable in Tris-HCl buffer solution (pH 7.4) during the observation time. The results suggest that **10a**–**10f** were responsible for the antitumor activity.

#### 2.2.2. In Vitro Cytotoxic Activity

All target compounds **10a**–**10f** were screened for their antitumor activities in vitro against tumor cell lines BGC-823, BEL-7402, MCF-7, and A549 with 5-FU as a positive control. Additionally, HL-7702 was chosen to evaluate the safety of **10a**–**10f** against normal human cell lines. The IC_50_ values of preliminary biological evaluation were summarized in Table 1 and Figure 2. The results showed that compounds **10a**–**10f** exhibited potent cytotoxic activity against selected tumor cell lines and less cytotoxicity toward normal cells HL-7702, which indicated that they all had broad-spectrum antitumor activity. Compounds **10a** and **10f** were the most potent compounds against BGC-823 and BEL-7402. Moreover, neither of them had cytotoxicity toward normal cells HL-7702. For **10a**, the IC_50_ values against BGC-823 and BEL-7402 cells were 9.00 µM and 6.70 µM, respectively, which was better than 5-FU (IC_50_ = 15.18 μM and 15.81 μM). Compound **10f** also displayed higher antitumor activity than 5-FU with IC_50_ values of 7.66 µM and 7.89 µM. Compared with some reported hydrazone derivatives, such as naproxen hydrazide-hydrazone (IC_50_ = 22.42 µM) [29], furanonyl sulfonyl hydrazone (IC_50_ = 14.35 µM) [30], and phenylhydrazone derivative (IC_50_ = 10.57 µM) [31], **10a** and **10f** showed better antitumor activity against MCF-7. The results revealed that **10a** and **10f** are promising antitumor agents. Thus, both of them were selected for further antitumor mechanism studies.

The preliminary structure–activity relationships of **10a**–**10f** were explored. The electronic effect had a remarkable influence on antitumor activity; thus, the effect of substituent (–Br, –H, and –CH_3_) on pyridine moiety on antitumor activity was discussed. It is evident that the antitumor activity of **10a**–**10c** against BGC-823, BEL-7402, and A549 cell lines all followed the descending order **10a** > **10b** > **10c.** The bromine-substituted compound **10a** showed the best antitumor activity, which indicated that the anticancer activity was prone to being sensitive to the electron-pulling substituent in the molecule. It is worth mentioning that the introduction of electron-pulling substituent can really improve the bioactivity of many compounds [32,33,34]. DNA is an important target for many small-molecule agents. Generally speaking, increasing the molecular plane of aromatic ring can enhance the binding ability of small molecule with DNA. Hence, **10d**–**10f** were specifically designed and synthesized to investigate the effect of molecular plane on antitumor activity. It was expected that the introduction of phenyl on pyrimidine skeleton can significantly enhance antitumor activity for the increase of conjugate structure. Regrettably, it was found that the introduction of phenyl did not significantly improve the antitumor activity, compared with the above series without phenyl (**10a**–**10f**). Perhaps, the introduction of phenyl also increased the molecules’ rigidity and steric hindrance, which is not beneficial to antitumor activity. Interestingly, when phenyl was introduced, the order of antitumor activity was completely reversed compared with **10a**–**10c** and **10d**–**10f** against BGC-823, BEL-7402, and MCF-7 cell lines all followed the descending order **10f** > **10e** > **10d;** the methyl-substituted compound **10f** showed the best antitumor activity, which means that the electron-pushing substituent is more beneficial when phenyl is on the pyrimidine skeleton. It should be mentioned that the addition of a phenyl group to the pyrimidine skeleton significantly reduces the toxicity to normal cells HL-7702.

### 2.3. Apoptosis Detection Analysis

#### 2.3.1. The Effect on Tumor Cell Morphology

AO/EB double staining is one of the simplest methods to investigate cell apoptosis. AO, as a vital dye showing green fluorescence, can stain viable cells, while EB stains only apoptotic or necrotic cells which have lost their membrane integrity and show orange or red fluorescence. Therefore, AO/EB double staining can be used to evaluate the morphology of living and dead cells (apoptotic and necrotic). The morphological changes of AO/EB dual-stained BGC-823 cells treated with varying concentrations of **10a** and **10f** (0 μM for control, 0.5 IC_50_, IC_50_, and 1.5 IC_50_) are shown in Figure 3. A normal morphology and homogeneous green fluorescence were observed in the control group. However, an increasing number of cells with apoptotic features such as nuclear fragmentation and chromatin condensation were observed in the test group in a dose-dependent manner. The results preliminarily indicate that both **10a** and **10f** can induce apoptosis of BGC-823 cells in a dose-dependent manner.

#### 2.3.2. Apoptosis Detection

Induction of apoptosis is considered to be one of the pathways for many antitumor agents. To further confirm whether **10a** and **10f** can induce apoptosis of BGC-823 cells, an Annexin V–FITC/PI kit was used to detect the degree of apoptosis by flow cytometry. As shown in Figure 4, compared with the control group, after BGC-823 cells were incubated with **10a** or **10f** gradient concentrations (0.5IC_50_, IC_50_, and 1.5IC_50_) for 48 h, the proportion of apoptotic cells increased in a concentration-dependent manner. As shown in Figure 4a,c, after BGC-823 cells were treated with **10a** (5 μM, 9 μM, and 13 μM) for 48 h, the percentage of the Q3 zone was significantly increased by 49.7%, 55.1%, and 60.5%, respectively. Similarly, when BGC-823 cells were incubated with **10f** (4 μM, 8 μM, and 12 μM), the proportion of Q3 zone was also remarkably increased (Figure 4b,c) by 44.3%, 54.8%, and 59.1%, respectively. The Q3 zone represents the early apoptotic cells. The above data revealed that both **10a** and **10f** could effectively induce early apoptosis in BGC-823 cells in a concentration-dependent manner. The results are consistent with that of AO/EB double staining.

#### 2.3.3. ROS Cell Fluorescence Intensity

ROS are key substances in the process of apoptosis. Excess ROS can attack the mitochondrial membrane, leading to the release of cytochrome C and apoptosis-inducing factors [35,36]. 2,7-Dichlorodihydrofluorescein diacetate (DCFH-DA, ROS Assay Kit) was used to evaluate the ROS production ability of **10f** against BGC-823 cells. As shown in Figure 5, the fluorescence intensity of the control group was zero, while there was also no significant intensity increase in the Rosup group. Figure 5a shows that the number of cells and intensity of green fluorescence increased with the increase in **10f** (4 μM and 8 μM). Meanwhile, Figure 5b revealed that the ROS generation increased when BGC-823 cells were treated with **10f** (4 μM and 8 μM); the fluorescence intensities were 27.6 and 32.2, respectively. The results indicated that **10f** can enhance the level of the intracellular ROS and facilitate cell apoptosis; this phenomenon has also been widely reported in the literature [37,38].

### 2.4. DNA-Binding Studies

Targeted therapy is an important mechanism of antitumor drugs. The interaction of small molecules with DNA is the common mechanism of antitumor effect [39]. There are three main noncovalent interaction modes of small molecules with DNA [40,41,42,43,44], which are electrostatic interaction, groove binding, and intercalation. The commonly used detection methods are spectrum, viscometry, and gel electrophoresis [45,46]. Spectral detection is the most widely used method because of speed and convenience. The interaction modes of small molecules with DNA can be determined by the changes of peak intensity and peak position.

#### 2.4.1. UV/Visible Spectral Analysis

Electron absorption spectroscopy is commonly used to examine the binding mode of small molecules with DNA [47]. The absorption spectra of **10a** and **10f** in the absence and presence of CT-DNA at varying concentrations were measured (Figure 6a,b). As shown, the absorption bands of **10a** and **10f** were similar and affected significantly. Sharp hyperchromic effects occurred at 205 nm and 260 nm with increasing concentration of CT-DNA; meanwhile, no apparent wavelength shift was observed, which suggested that interactions of **10a** and **10f** with CT-DNA were nonclassical intercalations, such as groove binding and partial intercalation mode. Hyperchromism has also been observed in other compounds [48,49,50], which is presumed to be due to the damage of the DNA double helix, whereby **10a** and **10f** are maybe embedded between DNA double helix base pairs, resulting in changes in the base-stacking force [51,52,53]. To investigate the binding affinity of **10a** and **10f** with CT-DNA, the binding constant (*K*_b_) was calculated according to the equation described in Section 3. The *K*_b_ values obtained for **10a** and **10f** were 4.44 × 10^6^ M^−1^ and 4.59 × 10^6^ M^−1^ (Figure 6c,d), which displayed a higher affinity of **10a** and **10f** with DNA. Although the *K*_b_ values for **10a** and **10f** were less than that of classical intercalator EB whose binding constant has been reported to be in the order of 10^7^ M^−1^ [54], they were still within the range of intercalation mode (*K*_b_ = 1 × 10^5^–1 × 10^11^ M^−1^) [54,55,56]. The absorption spectra results revealed that **10a** and **10f** probably interacted with DNA via groove binding and partial intercalation. The *K*_b_ values for **10a** and **10f** also imply that the binding affinity of **10f** with DNA was slightly stronger than that of **10a**, which might be attributed to the larger planar aromatic structure. In order to further confirm the binding mode of **10a** and **10f** with DNA, CD spectra and molecular docking were also performed.

#### 2.4.2. Circular Dichroism Spectral Analysis

The CD spectrum is a powerful tool for monitoring structural changes in DNA. The CD spectrum is extremely sensitive to conformational changes of DNA, especially in the range of 180–320 nm [57]. Binding of small molecules with DNA can change the DNA configuration, which will cause changes in the CD spectra. Electrostatic interaction and groove binding of small molecules usually cause no change or perturbation in the CD spectrum of DNA [28], while classical intercalation generally significantly changes the position and intensity of the positive and negative peaks of CD spectrum or leads to the generation of new induced circular dichroism (ICD) signals [58]. Figure 7 shows the CD spectra of CT-DNA in the absence and presence of **10a** or **10f**. Both the positive (279 nm) and the negative (247 nm) bands decreased in intensity after the addition of **10a** or **10f**, which suggested that the binding of **10a** and **10f** with CT-DNA induces certain conformational changes, unwinding the DNA helix [59]. These changes support the groove binding mode [60,61,62], which does not exclude a partial intercalation. It should be pointed out that **10f** caused a more obvious decrease in CD peak intensities, suggesting that **10f** interacted with DNA more strongly, which is consistent with the calculation results of UV/Vis spectra. A possible reason is that the introduction of phenyl increased the planarity of the molecule, strengthening DNA binding.

#### 2.4.3. Molecular Docking Study

To further verify the binding mode, AutoDock Tools-1.5.6 and PyMOL Molecular Graphics System were used to investigate the binding of **10a** or **10f** with DNA (PDB ID:1D18) through semiflexible molecular docking. The visualization is shown in Figure 8, Appendix A. It can be seen clearly that the interaction mode of **10a** and **10f** with DNA (PDB ID: 1D18) is not the classic parallel but oblique to the base pairs. As shown, the pyrimidine groups of **10a** and **10f** inserted into the base pairs of the DNA, while the pyridine ring on **10a** and **10f** extended toward the grooves in both sides of DNA. For compound **10f**, the phenyl on the pyrimidine skeleton extended toward the minor groove and formed π–π stacking (Figure 8b and Appendix A). Appendix A display that the interaction of **10a** and **10f** with DNA was groove binding and oblique intercalation. The docking poses showed that **10a** and **10f** stacked among the A, G, and C base pairs of the DNA intercalation site. The N atoms on hydrazone and pyridine ring formed hydrogen bonding with the DG-13 residue. Furthermore, for **10a**, the N and H atoms of hydrazone had hydrogen bonds with DG-4 and DG-13 residues; in the case of **10f**, the N and H atoms of hydrazone formed hydrogen bonding with DA-14 and DC-5 residues. The oblique docking position of **10a** and **10f** with DNA loosened the double helix of the DNA and reduced the stacking force of base pairs, which caused a decrease in the positive and negative peaks. The observed results were in agreement with CD spectral data.

Additionally, the binding energy (Δ*G*_b_^0^) for **10a** and **10f** interacting with DNA is listed in Appendix A. The Δ*G*_b_^0^ values for **10a** and **10f** were −8.0 kcal/mol and −8.1 kcal/mol, respectively. The negative values of Δ*G*_b_^0^ indicated that the binding of **10a** and **10f** with DNA was spontaneous. The Δ*G*_b_^0^ for **10f** was slightly less than that of **10a**, indicating stronger binding. The binding constants (*K*_b_) for **10a** and **10f** calculated according to this equation [63] ($\Delta G_{b}^{o}=-RT\ln K_{b}$) were 7.75 × 10^5^ M^−1^ and 9.18 × 10^5^ M^−1^ (Appendix A), which are also within the range of intercalation mode (*K*_b_ = 1 × 10^5^ to 1 × 10^11^ M^−1^) [54,55,56]. As a result, the binding of **10a** and **10f** with DNA involved groove binding and partial intercalation, which were consistent with UV/Vis and CD spectra.

Apoptosis can be activated by a variety of cellular signals, such as intracellular Ca^2+^ concentration, an increase in ROS levels, and DNA damage. Herein, ROS detection results showed that the intracellular ROS concentration increased with the increase in **10f** concentration, which can induce apoptosis. The results of UV/Vis, CD spectra, and molecular docking showed that DNA was an important target for the antitumor activity of these compounds, which may also cause cell apoptosis.

## 3. Materials and Methods

### 3.1. Materials

All biochemical reagents were purchased from Solarbio Science & Technology Co., Ltd. (Beijing, China), except fatal bovine serum (FBS), which was purchased from Lonsera (Uruguay). All cell lines (HL-7702, MCF-7, BGC-823, BEL-7402, and A549) were purchased from the Obio Technology (Shanghai) Corp., Ltd (Shanghai, China). The chemical raw materials used were obtained from J&K Scientific Ltd (Beijing, China).

### 3.2. Instrumentations

^1^H NMR and ^13^C NMR were measured using a nuclear magnetic resonance spectrometer (AVANCE III 500 MHz, Bruker, Germany). ESI-HRMS was recorded using high-resolution mass spectrometry (Thermo Scientific Q Exactive, Bruker, Germany). IR absorption spectra were recorded in the range of 4000–400 cm−1 on a WQF-510A FTIR spectrophotometer. UV/Vis spectra were recorded on a UV-3600i plus spectrophotometer (UV/Vis/NIR spectrophotometer, Shimadzu, Japan). CD spectra were recorded by circular dichromatic spectrometer (Chirascan/V100, Applied Photophysics, UK). In the MTT assay, the absorbance of each compound was measured at 570 nm using Multiskan GO (Thermofisher, Thermo Fisher Scientific, Waltham, MA, USA). AO/EB staining was performed under an inverted fluorescence microscope (DMIL LED, Leica, Germany). Cell apoptosis and ROS intensity were detected using the flow cytometry system (Beckman coulter CytoFLEX, Beckman Coulter, Pasadena, CA, USA). In addition, ROS intensity was also detected using an inverted fluorescence microscope (DMIL LED, Leica, Germany).

### 3.3. Synthesis

4,6-Dihydrazinylpyrimidine (**9a**) and 4,6-dihydrazinyl-2-phenylpyrimidine (**9b**) were the intermediates for the synthesis of target compounds **10a**–**10f**. The synthesis procedures of **9a** and **9b** are similar; they were prepared by using 4,6-dichloropyrimidine (**8a**) and 4,6-dichloro-2-phenylpyrimidine (**8b**) as starting materials in a reaction with excess hydrazine hydrate. Hence, the synthesis process of **9a** is described here.

The detailed synthesis procedure of **9a** was as follows: **8a** (15 mmol, 2.235 g) was dispersed in 15 mL hydrazine hydrate and stirred at room temperature for 1 h. Then, the mixture was heated to 60 °C to react continuously, and white solid gradually precipitated during the reaction. The reaction progress was monitored by thin-layer chromatography (TLC). After reaction was completed, the mixture was cooled overnight in a refrigerator at 4 °C to precipitate completely. Then, the precipitate was filtered and washed with mixed solvent of ethanol and water several times to obtain a white solid, which was named 4,6-dihydrazinylpyrimidine (**9a**), with a yield of ca. 72.0%. 4,6-Dihydrazinyl-2-phenylpyrimidine (**9b**) was also obtained using the same method with a yield of ca. 55.5%.

#### General Procedure for the Synthesis of Compounds **10a**–**10f**

The synthesis procedures of **10a**–**10f** were similar, as represented in Figure 1; therefore, only the synthesis of **10a** is introduced in detail.

4,6-Bis(2-((*E*)-(6-bromopyridin-2-yl)methylene)hydrazinyl)pyrimidine (**10a**). A mixture of **9a** (2 mmol, 280 mg) and 6-bromo-2-pyridinecarboxaldehyde (4.2 mmol, 781 mg) in 40 mL ethanol solvent was stirred at room temperature for 2 h, and then refluxed at 78 °C until the reaction was finished. The white solid gradually precipitated during the reaction, and the reaction was monitored by TLC. After the reaction was completed, the resulting solid was collected by filtration, washed with ethanol several times, and purified by column chromatography (petroleum ether: ethyl acetate = 1:1) to obtain target compound **10a**, with a yield of ca. 84.6%. ^1^H NMR (500 MHz, DMSO-*d_6_*) δ 11.58 (s, 2H, NH), 8.27 (s, 1H, N–CH=N, CH_pyrimidine_), 8.07 (s, 2H, CH=N), 7.99 (d, *J* = 7.5 Hz, 2H, CH_pyridine_), 7.83 (t, *J* = 7.8 Hz, 2H, CH_pyridine_), 7.62 (d, *J* = 8.2 Hz, 2H, CH_pyridine_), 6.94 (s, 1H, CH_pyrimidine_). ^13^C NMR (101 MHz, DMSO-*d_6_*) δ 161.99, 158.39, 155.29, 141.47, 140.76, 140.54 (CH=N), 128.00, 119.35, 82.78. ESI-HRMS: *m/z* calculated for [C_16_H_12_Br_2_N_8_ + H]^+^: 476.96095, found: 476.95822. IR (cm^−1^, KBr): 3193, 1582, 1569, 1544, 1454, 1432, 1419, 1394, 1201, 1160, 1137, 1116, 792.

4,6-Bis(2-((*E*)-pyridin-2-ylmethylene)hydrazinyl)pyrimidine (**10b**). Yellow solid; yield: ca. 68.8%. ^1^H NMR (500 MHz, DMSO-*d_6_*) δ 11.40 (s, 2H, NH), 8.58 (d, *J* = 6.3 Hz, 2H, CH_pyridine_), 8.22 (d, *J* = 1.0 Hz, 2H, CH_pyridine_), 8.16 (s, 1H, N–CH=N, CH_pyrimidine_), 7.99 (d, *J* = 8.0 Hz, 2H, CH_pyridine_), 7.89 (s, 2H, CH=N), 7.36 (d, *J* = 7.4 Hz, 2H, CH_pyridine_), 6.90 (s, 1H, CH_pyrimidine_). ^13^C NMR (101 MHz, DMSO-*d_6_*) δ 162.08, 158.33, 154.07, 149.91, 142.88, 137.30 (CH=N), 124.05, 119.85, 82.30. ESI-HRMS: *m/z* calculated for [C_16_H_14_N_8_ + H]^+^: 319.14197, found: 319.14069. IR (cm^−1^, KBr): 3189, 1598, 1577, 1548, 1467, 1454, 1432, 1413, 1203, 1135, 985, 777.

4,6-Bis(2-((*E*)-(6-methylpyridin-2-yl)methylene)hydrazinyl)pyrimidine (**10c**). Yellow solid; yield: ca. 87.5%. ^1^H NMR (500 MHz, DMSO-*d_6_*) δ 11.32 (s, 2H, NH), 8.15 (s, 1H, N–CH=N, CH_pyrimidine_), 8.04 (s, 2H, CH=N), 7.70 (d, *J* = 7.5 Hz, 4H, CH_pyridine_), 7.16 (d, *J* = 6.2 Hz, 2H, CH_pyridine_), 6.83 (s, 1H, CH_pyrimidine_), 2.42 (s, 6H, –CH_3_). ^13^C NMR (101 MHz, DMSO-*d_6_*) δ 161.97, 158.01, 153.27, 142.82, 137.26 (CH=N), 123.37, 117.00, 82.38, 24.08 (–CH_3_). ESI-HRMS: *m/z* calculated for [C_18_H_18_N_8_ + H]^+^: 347.17327, found: 347.17218. IR (cm^−1^, KBr): 3183, 2969, 1569, 1544, 1450, 1423, 1403, 1199, 1157, 1135, 985.

4,6-Bis(2-((*E*)-(6-bromopyridin-2-yl)methylene)hydrazinyl)-2-phenylpyrimidine (**10d**). White solid; yield: ca. 68.8%. ^1^H NMR (500 MHz, DMSO-*d_6_*) δ 11.59 (s, 2H, NH), 8.29–8.20 (dd, *J* = 7.8 Hz, 2H, CH_pyridine_), 8.05 (s, 2H, CH=N), 7.95 (d, *J* = 7.7 Hz, 2H, CH_pyridine_), 7.76 (t, *J* = 7.8 Hz, 2H, CH_pyridine_), 7.54 (d, *J* = 7.8 Hz, 2H, CH_benzene_), 7.48–7.40 (m, *J* = 7.3 Hz, 3H, CH_benzene_), 6.87 (s, 1H, CH_pyrimidine_). ^13^C NMR (101 MHz, DMSO-*d_6_*) δ 163.34, 162.62, 155.36, 141.47, 140.67, 140.56, 138.04 (CH=N), 130.91, 128.79, 128.07, 127.98, 119.33, 81.43. ESI-HRMS: *m/z* calculated for [C_22_H_16_Br_2_N_8_ + H]^+^: 552.99225, found: 552.99109. IR (cm^−1^, KBr): 3299, 1592, 1558, 1540, 1525, 1434, 1413, 1394, 1378, 1195, 1170, 891, 786.

4,6-Bis(2-((*E*)-pyridin-2-ylmethylene)hydrazinyl)-2-phenylpyrimidine (**10e**). Yellow solid; yield: ca. nearly 65.5%. ^1^H NMR (500 MHz, DMSO-*d_6_*) δ 11.51 (s, 2H, NH), 8.61 (d, *J* = 4.2 Hz, 2H, CH_pyridine_), 8.34 (dd, *J* = 6.6, 3.1 Hz, 2H, CH_pyridine_), 8.24 (s, 2H, CH=N), 8.04 (d, *J* = 8.0 Hz, 2H, CH_benzene_), 7.92 (td, *J* = 7.7, 1.5 Hz, 2H, CH_pyridine_), 7.58–7.47 (m, *J* = 7.3 Hz, 3H, CH_benzene_), 7.43–7.33 (m, *J* = 7.6 Hz, 2H, CH_pyridine_), 6.93 (s, 1H, CH_pyrimidine_). ^13^C NMR (101 MHz, DMSO-*d_6_*) δ 163.23, 162.71, 154.12, 149.90, 142.79, 138.19 (CH=N), 137.34, 130.81, 128.75, 128.06, 124.05, 119.87, 81.00. ESI-HRMS: *m/z* calculated for [C_22_H_18_N_8_ + H]^+^: 395.17327, found: 395.17139. IR (cm^−1^, KBr): 3199, 1592, 1558, 1467, 1432, 1417, 1392, 1201, 1168, 1147, 1106, 701.

4,6-Bis(2-((*E*)-(6-methylpyridin-2-yl)methylene)hydrazinyl)-2-phenylpyrimidine (**10f**). Yellow solid; yield: ca. 55.5%. ^1^H NMR (500 MHz, DMSO-*d_6_*) δ 11.44 (s, 2H, NH), 8.33–8.26 (dd, *J* = 6.6, 3.1 Hz, 2H, CH_pyridine_), 8.14 (s, 2H, CH=N), 7.81–7.72 (m, 7.6 Hz, 4H, CH_pyridine_), 7.50–7.43 (t, *J* = 7.26 Hz, 3H, CH_benzene_), 7.20 (d, *J* = 7.3 Hz, 2H, CH_benzene_), 6.88 (s, 1H, CH_pyrimidine_), 2.47 (s, 6H, –CH_3_). ^13^C NMR (101 MHz, DMSO-*d_6_*) δ 163.28, 162.74, 158.21, 153.51, 142.93, 138.18 (CH=N), 137.57, 130.77, 128.74, 128.06, 123.33, 116.92, 81.16, 24.69 (–CH_3_). ESI-HRMS: *m/z* calculated for [C_24_H_22_N_8_ + H]^+^: 423.20457, found: 423.20343. IR (cm^−1^, KBr): 3191, 3033, 1596, 1560, 1538, 1454, 1417, 1386, 1191, 1160, 1139, 689.

### 3.4. Evaluation of Antitumor Activity In Vitro

#### 3.4.1. Stability Studies

The stability of **10a**–**10f** was the basis for the evaluation of antitumor activity, which was tested by UV/Vis. The stock solution of **10a**–**10f** (10^−3^ M) was prepared with DMSO. Considering the environment of cytotoxicity assay, the time-dependent UV/Vis absorption spectra of **10a**–**10f** (0.1 µM) at 0 h, 24 h, and 48 h were recorded in Tris-HCl buffer (0.05 M Tris-HCl/0.1 M NaCl, pH 7.4), respectively. The stability of **10a**–**10f** was determined by the changes in the position and intensity of the spectral absorption peaks.

#### 3.4.2. MTT Assay

The MTT method is often used to detect the cytotoxicity of compounds in vitro [64]. The cytotoxicity of **10a**–**10f** was evaluated by MTT assay against BGC-823, BEL-7402, MCF-7, and A549 tumor cells. HL-7702 was selected to evaluate the safety of **10a**–**10f** against human normal cells. 5-FU was also evaluated as a positive control. The well-grown cells were inoculated in 96-well plates at a density of 3000 cells/well and incubated in 37 °C for 24 h. The negative control group, blank control group, and administration group were set up. The cells were incubated with a gradient concentration (1.25, 2.5, 5, 10, and 20 μg/mL) of samples or 5-FU for 48 h. Then, 20 μL MTT solution (5 mg/mL) was added to each well. After the cells were incubated for 4 h, 150 μL DMSO was added to each well before testing. The absorbance was measured at 570 nm with a microplate reader. The inhibition rate and final IC_50_ values were calculated according to the following formula. The values shown are the average of at least three parallel trials.
Inhibition rate (%) =1 − (OD_sample_ − OD_blank_)/(OD_negative_ − OD_blank_) × 100%.

### 3.5. Apoptosis Detection

#### 3.5.1. AO/EB Double Staining

BGC-823 cells were inoculated in six-well plates and incubated at 37 °C for 24 h, and then incubated continuously for another 24 h with a varying concentration of **10a** or **10f** (0.5IC_50_, IC_50_, and 1.5IC_50_). AO (500 μL, 1 mg/mL) and EB (500 μL, 1 mg/mL) were mixed, and then 4 mL PBS was added to the above solution to form AO/EB dye (0.1 mg/mL). AO/EB dye (100 μL) was added to each well. After incubating for 3–5 min and washing twice with cold PBS, BGC-823 cells were observed immediately under an inverted fluorescence microscope.

#### 3.5.2. Flow Cytometry

The Annexin V–FITC/PI Kit was used to distinguish different stages of apoptosis [65,66]. BGC-823 cells were inoculated in six-well plates at 37 °C for 24 h, and then incubated for another 48 h with a varying concentration of **10a** or **10f** (0.5 IC_50_, IC_50_, and 1.5 IC_50_). The cells were trypsinized (without EDTA), and no fewer than 10^6^ cells were collected. According to the instructions, the Annexin V–FITC/PI kit was used to monitor induced apoptosis.

#### 3.5.3. ROS Detection

BGC-823 cells were inoculated in six-well plates at 37 °C for 24 h, and then incubated for another 24 h with a varying concentration of **10f** (0.5IC_50_, IC_50_, and 1.5IC_50_). According to the instructions of ROS Detection Kit (Solarbio, Beijing, China), in situ probe loading was adopted. ROS intensity changes were determined by inverted fluorescence microscopy and flow cytometry. The whole experiment was carried out under dark conditions.

### 3.6. DNA-Binding Study

#### 3.6.1. UV/Vis Spectra

The binding patterns of **10a** and **10f** with CT-DNA were measured by UV/Vis spectroscopy in Tris-HCl buffer (0.05 M Tris-HCl/0.1 M NaCl, pH 7.4). The stock solution concentrations of both **10a** and **10f** in DMSO were 10^−3^ M. CT-DNA was dissolved in Tris-HCl buffer and stored at 4 °C for use. The stock solution concentration of CT-DNA was determined by UV/Vis absorbance at 260 nm with the extinction coefficient 6600 M^−1^·cm^−1^ [67]. Tris-HCl buffer was used as a blank to make preliminary adjustments. Absorbance titration experiments were performed at room temperature with the concentrations of **10a** and **10f** at 0.1 µM, varying the concentrations of CT-DNA (0.2, 0.4, 0.6, 0.8, and 1 µM). The solution was allowed to equilibrate for 2–3 h before measurement. UV/Vis absorption spectra were recorded in the range of 200–400 nm. The binding constant was calculated according to the following equation [54]:$$\left[\mathrm{DNA}\right]-(\varepsilon_{a}-\varepsilon_{f})=\left[\mathrm{DNA}\right]/\left(\varepsilon_{b}-\varepsilon_{f}\right)+1/K_{b}(\varepsilon_{b}-\varepsilon_{f}),$$
where *ε*_a_ corresponds to the observed extinction coefficient (*A*/[DNA]), *ε*_f_ corresponds to the extinction coefficient of the free compound, and *ε*_b_ is the extinction coefficient of the compound when fully bound to CT-DNA. [DNA] is the concentration of CT-DNA added to the compound. The *K*_b_ values of **10a** and **10f** were calculated from the slope to intercept ratio, which was obtained by plotting a graph of [DNA]/(*ε*_a_ − *ε*_f_) versus [DNA].

#### 3.6.2. Circular Dichroism Spectra

The CD spectra of CT-DNA in the absence and presence of **10a** or **10f** were measured after equilibrating for 1 h in a 1 mm diameter quartz colorimeter at room temperature. The concentration of CT-DNA was fixed at 600 μM in each sample. Both **10a** and **10f** were added to the above CT-DNA solution at a concentration of 100 μM, respectively. The scanning speed was 200 nm/min. The spectra were scanned on an average of six times in the range of 200–400 nm. All measurements were performed in 0.05 M Tris-HCl buffer (pH 7.4).

#### 3.6.3. Molecular Docking

Molecular docking of **10a** and **10f** with DNA was carried out. The molecular structures of **10a** and **10f** were plotted by ChemDraw and converted into MOL2 format by Chem3D. DNA (PDB ID: 1D18) was selected from the Protein database (http://www.rcsb.org./pdb, accessed on 23 November 2022). AutoDock Tools-1.5.6 software was used to process the molecular structures of **10a**, **10f**, and DNA, which were saved in PDBQT format. The binding energy ($\Delta G_{b}^{o}$) of **10a** and **10f** interacting with DNA was calculated. The docking posture was visualized using the PyMOL Molecular Graphics System.

## 4. Conclusions

In summary, a series of dihydrazone pyrimidine derivatives were synthesized successfully, characterized, and evaluated biologically. MTT results showed that all of them exhibited potential cytotoxic activity against selected tumor cells and less cytotoxicity toward normal cells HL-7702, indicating a good selectivity toward tumor cells. Among them, **10a** and **10f** displayed more potent cytotoxic activity against BGC-823 and BEL-7402 cells than 5-FU. Mechanism studies showed that **10a** and **10f** could induce early apoptosis in BGC-823 cells. Compounds **10a** and **10f** bind with DNA via groove binding and partial intercalation. Our work suggests that dihydrazone pyrimidine is a promising chemical skeleton and could be potentially utilized for designing more excellent antitumor agents.

## Data Availability

Not applicable.

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
