# Peer review of "Synthesis, Biological Evaluation, DNA Binding, and Molecular Docking of Hybrid 4,6-Dihydrazone Pyrimidine Derivatives as Antitumor Agents"

_molecules, 2022, doi:10.3390/molecules28010187_

Round 1

Reviewer 1 Report

The manuscript by Hai-Rong Lan and co-workers, entitled“Synthesis, Biological Evaluation, DNA-Binding and Molecular Docking of Novel Hybrid 4,6-Dihydrazone Pyrimidine Derivatives as Antitumor Agents”describes the synthesis, biological evaluation and DNA-Binding properties of a series of 4,6-dihydrazone pyrimidine derivatives containing pyridine moiety. The experiments in this paper were well organized and conducted systematically, and the results were clearly described. This work is of interest to those working in the field of medicinal chemistry. But I have a few questions for the authors to answer. After addressing these  issues, I would recommend the acceptance of the manuscript for publication in Molecules. 

1. What is the purity of the tested compounds? For biological experiments, the purity of a compound is crucial.

2. The authors suggest that the target of these compounds is DNA, so do they have an effect on the cell cycle?

3. Do these compounds affect mitochondrial functions? Such as mitochondrial membrane potential. Many anticancer drugs induce apoptosis accompanied by mitochondrial dysfunction.

Author Response

We are grateful to you for the positive comments and suggestions on our manuscript. We have carefully checked and accordingly corrected as best we can in the revised manuscript. Thanks for your careful reviewing and good advice on our manuscript.

Reviewer #1:

Comments and Suggestions for Authors

The manuscript by Hai-Rong Lan and co-workers, entitled“Synthesis, Biological Evaluation, DNA-Binding and Molecular Docking of Novel Hybrid 4,6-Dihydrazone Pyrimidine Derivatives as Antitumor Agents” describes the synthesis, biological evaluation and DNA-Binding properties of a series of 4,6-dihydrazone pyrimidine derivatives containing pyridine moiety. The experiments in this paper were well organized and conducted systematically, and the results were clearly described. This work is of interest to those working in the field of medicinal chemistry. But I have a few questions for the authors to answer. After addressing these issues, I would recommend the acceptance of the manuscript for publication in Molecules.  

Response: We are grateful to the reviewer for the positive comments and suggestions on our manuscript. We have carefully checked and accordingly corrected as best we can in the revised manuscript. Thanks for your careful reviewing and good advice on our manuscript.

  1. What is the purity of the tested compounds? For biological experiments, the purity of a compound is crucial.

Response: Thanks for your careful reviewing. The tested compounds were purified by column chromatography before being used for biological evaluation. The purification process of these compounds was described in synthesis section, and the purity of them was also confirmed by 1H NMR, 13C NMR and HRMS. What’s more, the 13C NMR spectra have been accordingly added in the revised manuscript.

  1. The authors suggest that the target of these compounds is DNA, so do they have an effect on the cell cycle?

Response: Thanks for your good advice. Indeed, many apoptosis-inducing drugs can cause cell cycle arrest. This research is well worthy of investigation. We will perform our future studies along this line. Thank you very much again.

  1. Do these compounds affect mitochondrial functions? Such as mitochondrial membrance potential. Many anticancer drugs induce apoptosis accompanied by mitochondrial dysfunction.

Response: Thanks for your good advice. Mitochondrial membrance potential is one of the indexes to evaluate the mechanism of apoptosis and worth studying. We will perform our future studies along this line. Thank you again for your good advice.

Thank you again for the positive comments and suggestions on our manuscript.

Reviewer 2 Report

1. The compound 10e reported in the article are not novel compounds and has been reported in the literature (Ramírez, J., Stadler, A.-M., Harrowfield, J.M., Brelot, L., Huuskonen, J., Rissanen, K., Allouche, L. and Lehn, J.-M. Z. anorg. allg. Chem., 2007, 633: 2435-2444. https://doi.org/10.1002/zaac.200700354). Therefore, the authors need to search for more literature, and it is necessary to add an explanation of which compounds are new and which are already reported in the Introduction.

2. In the Abstract, “…a novel series of 4,6-dihydrazone pyrimidine derivatives containing pyridine moiety was synthesized….”. In the Introduction, “….a series of novel 4,6-dihydrazone pyrimidine derivatives (10a~10f)….”, the “novel” is not used properly in this paper.

3. All compounds were not characterized by 13C NMR. Please provide the 13C NMR.

4. In the Scheme 1, the 10e and 10f are not labeled.

5. What is the yield of the compound 9a and 9b? In the synthesis procedures of 9a and 9b, the excess of hydrazine hydrate was used in this reaction, and 8a and 8b as starting materials contains two halogens, is it possible that the polymerization reaction may occur so that the yield decreases?

6. In 4.3. Synthesis, “1H NMR (500 MHz, DMSO)….”, this should be DMSO-d6. Besides, the ratio of the number of hydrogen protons does not agree with the structure of the compound, nor does it agree with the most simple ratio in the spectrum (Supplementary materials).

7. In this work, the authors investigated the effects of compounds on apoptosis, ROS, and the interactions with CT-DNA. Whether there is some connection between compound-induced apoptosis, ROS activation, and CT-DNA binding. It is necessary to be explained this connection in the discussion section.

8. In line 205, the authors describe “…BGC-823 cells in a dose-dependent manner”. I think this is an inappropriate statement here because the authors did not make a statistical analysis.

9. The authors should carefully check the figure and figure notes. For example, in figure 3, is the magnification of the control group in figure (a) consistent with that of the drug test group? According to the pictures provided by the authors, the nuclei of the cells are significantly enlarged after the action of the compounds? This is obviously inconsistent with the morphology of apoptosis. Besides, the magnification in Figures (a) and (b) should keep consistent.

Author Response

We are grateful to you for the positive comments and suggestions on our manuscript. We have carefully checked and accordingly corrected as best we can in the revised manuscript. Thanks for your careful reviewing and good advice on our manuscript.

Reviewer #2:

Comments to the Author

  1. The compound 10e reported in the article are not novel compounds and has been reported in the literature (Ramirez, J., Stadler, A.-M., Harrowfield, J.M., Brelot, L., Huuskonen, J., Rissanen, K., Allouche, L. and Lehn, J.-M. Z. anorg. Allg. Chem. 2007, 633, 2435-2444. https://doi.org/1002/zaac.200700354.). Therefore, the authors need to search for more literature, and it is necessary to add an explanation of which compounds are new and which are ready reported in the introduction.

Response: Thanks for your reminding. We have accordingly corrected. We have deleted the word "Novel" from the title for rigor, noted the related compound in the introduction and also added the relevant literature in References.

  1. In the Abstract, “…a novel series of 4,6-dihydrazone pyrimidine derivatives containing pyridine moiety was synthesized…”. In the introduction, “…a series of novel 4,6-dihydrazone pyrimidine derivatives (10a~10f)... ”, the “novel” is not used properly in this paper.

Response: We are grateful to you for your suggestions. We have carefully checked and deleted the word "novel".

  1. All compounds were not characterized by 13C NMR. Please provide the 13

Response: Thanks for your good advice. The 13C NMR spectra of all the compounds were measured and also added in Supplementary materials.

  1. In the Scheme 1, the 10e and 10f are not labeled.

Response: We are sorry for the neglect. We have added the labels to the Scheme 1.

  1. What is the yield of the compound 9a and 9b? In the synthesis procedures of 9a and 9b, the excess of hydrazine hydrate was used in this reaction, and 8a and 8b as starting materials contains two halogens, is it possible that the polymerization reaction may occur so that the yield decreases?

Response: Thanks for your careful reviewing. The yield of compound 9a was 72% and that of compound 9b was 55.5%, which have been added in the revised manuscript. The polymerization reaction was not detected in this system. The excess of hydrazine hydrate was used as both reactant and solvent.

  1. In 4.3. Synthesis,“1H NMR (500MHZ, DMSO).... ”, this should be DMSO-d6. Besides, the ratio of the number of hydrogen protons does not agree with the structure of the compound, nor does it agree with the most simple ratio in the spectrum (Supplementary materials).

Response: Thanks for your reminding. We have accordingly corrected DMSO to DMSO-d6. Additionally, we reconfirmed the data. The previous statements are indeed confusing. The series of these compounds are symmetric. The previous data was processed by the half of the molecule. We have reattributed the data by a whole molecule and updated the data in the revised manuscript.

  1. In this work, the authors investigated the effects of compounds on apoptosis, ROS, and the interactions with CT-DNA. Whether there is some connecttion between compound-induced apoptosis, ROS activation, and CT-DNA binding. It is necessary to be explained this connection in the discussion section.

Response: We are grateful to you for your suggestions. Apoptosis can be activated by a variety of cellular signals, such as intracellular Ca2+ concentration, the increase in ROS levels and DNA damage. We have added the relevance of above three items in the revised manuscript.

  1. In line 205, the authors describe“...BGC-823 cells in a dose-dependent manner”. I think this is an inappropriate statement here because the authors did make a statistical analysis.

Response: We are grateful to you for your suggestions. In this section, BGC-823 cells were incubated with 10a or 10f gradient concentrations (0.5IC50, IC50 and 1.5IC50) for 48 h, the proportion of apoptotic cells increased in a concentration dependent manner. We have corrected "dose-dependent manner" to "concentration-dependent manner" in the revised manuscript.

  1. The authors should carefully check the figure and figure notes. For example, in figure 3, is the magnification of the control group in figure(a) consistent with that of the drug test group? According to the pictures provided by the authors, the nuciel of the cells are significantly enlarged after the action of the compounds? This is obviously inconsistent with the morphology of apoptosis. Besides, the magnification in figures (a) and (b) should keep consistent.

Response: We are grateful to you for your suggestions. We have carefully checked the figure and figure notes.We are very sorry for the neglect. The control group in Figure 3(a) has been replaced with images with the same magnification as the test group. We are very sorry that figures (a) and (b) are not in the same batch of experiments, and the images with the same magnification were not saved at that time. Scale bars were added to each image in the revised manuscript.

Thank you again for the positive comments and suggestions on our manuscript.

Round 2

Reviewer 2 Report

The author has revised the manuscript, which I think is acceptable